# The Clinical Use of Osteobiologic and Metallic Biomaterials in Orthopedic Surgery: The Present and the Future

**DOI:** 10.3390/ma16103633

**Published:** 2023-05-10

**Authors:** Sung-ryul Choi, Ji-won Kwon, Kyung-soo Suk, Hak-sun Kim, Seong-hwan Moon, Si-young Park, Byung Ho Lee

**Affiliations:** 1Department of Orthopedic Surgery, Spine and Spinal Cord Institute, Gangnam Severance Hospital, Yonsei University College of Medicine, Seoul 06273, Republic of Korea; srchoi1012@gmail.com (S.-r.C.);; 2Department of Orthopedic Surgery, Yonsei University College of Medicine, Seoul 03722, Republic of Korea

**Keywords:** osteobiology, biomaterials, polymers, ceramics, metals

## Abstract

As the area and range of surgical treatments in the orthopedic field have expanded, the development of biomaterials used for these treatments has also advanced. Biomaterials have osteobiologic properties, including osteogenicity, osteoconduction, and osteoinduction. Natural polymers, synthetic polymers, ceramics, and allograft-based substitutes can all be classified as biomaterials. Metallic implants are first-generation biomaterials that continue to be used and are constantly evolving. Metallic implants can be made from pure metals, such as cobalt, nickel, iron, or titanium, or from alloys, such as stainless steel, cobalt-based alloys, or titanium-based alloys. This review describes the fundamental characteristics of metals and biomaterials used in the orthopedic field and new developments in nanotechnology and 3D-printing technology. This overview discusses the biomaterials that clinicians commonly use. A complementary relationship between doctors and biomaterial scientists is likely to be necessary in the future.

## 1. Introduction

In the field of orthopedics, the application of surgical treatments to address degenerative diseases and trauma has been increasing gradually in many countries [1]. Orthopedic surgery has seen significant advances in the use of biomaterials for bone repair and reconstruction. Osteobiologic and metallic biomaterials, in particular, have gained widespread attention for their ability to promote bone healing and improve patient outcomes. Even within the field of orthopedic surgery, the mechanical properties required for each surgery are different, and biomaterials are different accordingly. For example, plates used in fracture surgery and stems used in arthroplasty surgery use metal, especially titanium alloys, which, compared to other options, have elastic moduli that are more similar to cortical bone. In addition, of all bearing surfaces now employed in orthopedic surgery, ceramic bearing surfaces have shown some of the best wear attributes (Figure 1) [2,3]. However, with so many types of biomaterials available, it can be challenging for clinicians and researchers to navigate the options and determine which materials are most suitable for their patients’ needs.

This narrative review journal seeks to offer a comprehensive overview of the therapeutic application of metallic and osteobiologic biomaterials in orthopedic surgery. The review draws upon existing literature to explore the properties and characteristics of various biomaterials, their clinical applications, and surgical techniques. The review also examines the outcomes of different biomaterials in orthopedic surgery and the potential for future developments in this field.

The review is organized into several sections, each covering a different aspect of the clinical use of osteobiologic and metallic biomaterials. These sections include a description of the many biomaterial types, their characteristics, and their clinical applications. The review also examines the surgical techniques used for implanting biomaterials and the potential complications that can arise. Additionally, the review explores the outcomes of different biomaterials, including their effectiveness in promoting bone healing and restoration of function.

There have been numerous studies about bone biomaterials. However, the majority of papers are written from the perspective of a material scientist and few cases that are written from the perspective of clinicians and used in actual clinical practice are available. Therefore, this review journal will be able to show the view of biomaterials from the perspective of a clinician to material scientists who research and develop biomaterials. The authors of this narrative review are experts in the field of biomaterials and orthopedic surgery, with extensive experience in clinical practice, research, and education. Through their collective expertise, they provide a comprehensive and authoritative overview of the current state of biomaterials in orthopedic surgery, with insights into the challenges and opportunities that lie ahead.

The low mechanical strength and brittleness of biomedical implants made of polymers and ceramics restricts their use in demanding working environments [4]. Therefore, it is currently important to produce prostheses primarily formed of metallic materials for orthopedic surgery.

The majority of implants—roughly 70–80%—are composed of metallic biomaterials. The metals 316L stainless steel, CoCrMo alloys, and Ti and its alloys, such as Ti6Al4V and NiTi alloys, are the most often utilized metals for medical purposes [5]. Metals often have a significantly higher modulus of elasticity than bone, which limits stress transfer to the bone. Because of the disproportionately large gap between the elastic modulus of metal and bone, the stress shielding effect might have negative result in bone remodeling. Few acceptable metal candidates are recommended for use as long-term implants because of the harmful effect the ions they produce have on the surrounding tissues, their weak resistance to wear and corrosion, and, as a result, their restricted biocompatibility. By alloying the basic metal with compounds that have no harmful effects on the body, these issues can be remedied. Corrosion, which is the degradation of a material as a result of an electrochemical attack in its surroundings, is one of the most important dangers that can be brought on by a metallic implant [5,6,7].

Overall, this narrative review journal provides a valuable resource for clinicians and researchers interested in exploring the potential of osteobiologic and metallic biomaterials in orthopedic surgery. It serves as a comprehensive reference guide for anyone seeking to navigate the options available and make informed decisions about the use of biomaterials for bone repair and reconstruction.

## 2. Osteobiology

Osteobiologic properties include osteogenicity, osteoconduction, and osteoinduction.

The term “osteogenicity” describes a scaffold’s or implant’s ability to promote the production of new bone in the absence of host cell invasion. Before being implanted, cells must be seeded onto the scaffold for it to be osteogenic [8,9]. Bones serve as the basis for our physical movement, supports our skeleton and protects our internal organs, houses the biological components necessary for hematopoiesis, traps hazardous metals (such as lead), and preserves the homeostasis of important electrolytes through calcium and phosphate ion storage. Additionally, bone is always undergoing chemical exchange and structural remodeling as a result of both internal mediators and external mechanical stresses. As a result, resorption and renewal occur continuously [10,11]. Bone has been described as the “ultimate smart material” in the past, which is most suitable; hence, any graft, scaffold, or implant used to replace missing or damaged bone tissue needs to have similar mechanical qualities.

Osteoconduction refers to the ability to promote the attachment of osteoblastic cells to a scaffold or implant on the surface and throughout the interior. In in vitro settings, osteoconduction refers to the ability to promote the attachment, migration, and proliferation of osteoblasts [9,12,13].

The term “osteoinduction” refers to the ability of a scaffold or implant to encourage the differentiation of mesenchymal stem cells down an osteoblastic lineage, ultimately resulting in the development of mineralized tissue. The ability to encourage the phenotypic transition from an early osteoblast to a mature osteoblast, followed by differentiation into an osteocyte, is referred to as osteoinduction [14].

## 3. The Generation of Biomaterials

In early stage, the most important property of an orthopedic device was biological inertness, which explains an absence of responsiveness to the biological environment. However, since bioinertness does not help bone healing, there have been developments of biomaterials that help bone healing for decades. Some level of reactivity is always expected in vivo because of oxidation or foreign body reaction; therefore, a low level of implant degradation may be acceptable if mechanical strength is not impaired and no harmful byproducts are produced [15].

The regulation of reactivity between implant materials and the surrounding biological environment can hasten the recovery process [16]. Dolcimascolo et al. [17] assert that three generations of biomaterials have been utilized in the past 60 years: bio-inert materials (first generation), bioactive or biodegradable materials (second generation), and current materials (third generation). In addition, recent studies show that biomimetic biomaterials were called fourth generation. The major requirements for first-generation implant materials were material stiffness and biological inertness [18]. A thin, acellular fibrous capsule serves as the interface between tissues and bioinert biomaterials, with only a slight adhesion between the implant and its host tissue. This adherence is crucial to the stability of the initial generation of biomaterials [19]. Between 1980 and 2000, second-generation materials were developed to resolve this issue. Second-generation implant materials are biodegradable, bioreabsorbable, or bioactive materials allow implants created with an in vivo disintegration rate so that the structure’s strength is maintained until the tissue-engineered transplant is fully remodeled and ultimately takes on its structural role [20]. Biodegradable materials that degrade gradually allow for the healing and regeneration of tissues near the implant [21]. Bioresorbable polymers are metabolized by the human body after implantation and include numerous substances, including polylactide (PLA), polyglycolide (PGA) polyhydroxyalkanoates(PHAs), polycaprolactone(PCL), and copolymers of PLA/PGA. With these materials, non-structural medication administration to resorbable screws and anchors can be achieved while also meeting mechanical performance and resorption rates. The particular property of second -generation of biomaterials, which enhance the biological response and the tissue/surface bonding, or that have ability to degrade while new tissue regenerates and heals, compared to first-generation is that they can interact with their biological surroundings. With increased reliability and decreased incidence of complications, their application has become widespread. In orthopedic surgery, clinicians use suture anchors for rotator cuff injury or ligament repair surgery. In animal studies, there have been attempts to use bioreabsorbable interbody spacers in the field of spine surgery. These spacers slowly disintegrate in a physiological environment, allowing a gradual transfer of loads to the healing tissue [22]. Thus, there is no longer a requirement for implant removal surgery, and the dangers of long-term issues with bioinert materials—such as stress shielding, migration, late infection, and interference with imaging tests such as CT and MRI—are diminished [23]. Instead of forming a fibrous capsule, bioactive materials have been extensively researched for their potential to induce mineral deposition in biological environments. Among these materials, bioactive glass has shown promise in precipitating hydroxyapatite upon contact with bodily fluids, making it a viable candidate for bone regeneration [24,25]. Because hydroxyapatite constitutes approximately 66% of bone diameter, bone bonding occurs through the creation of a surface layer of hydroxycarbonate apatite (HCA) that mimics the chemical and crystallographic characteristics of bone [26,27]. Biomaterials of the third generation possess the capability to elicit targeted cellular responses at the molecular scale. The design of third-generation materials aims to expedite the healing process by promoting molecular reactions with specific properties. In third-generation biomaterials, bioresorbable and bioactive materials are utilized to construct temporary three-dimensional porous structures that can trigger gene expression for the purpose of stimulating tissue regeneration. The ideas of bioactivity and biodegradability are united in these biomaterials, making this combination the one that characterizes the third-generation biomaterials most. Porous structures that concentrate on titanium and titanium alloys have been developed using metals. These biomaterials are intended to be transient, three-dimensional (3D), porous structures that can promote angiogenesis, nutrition uptake, and tissue regeneration, and are used in the fields of regenerative medicine, tissue transplantation, grafting, and tissue engineering [28,29]. The fourth generation of biomaterials, known as smart biomaterials, has the property of being able to mimic natural structures and mechanisms, repairing and regenerating damaged tissues by encouraging particular cell reactions. This generation has been made possible by advances in various fields. These materials are also known as “biomimetic biomaterials.” They can be temporary devices, such as plates or screws, or they might develop a permanent presence, such as prostheses. Although they are the most recent generation of biomaterials and offer some advantages over others, it is still impractical to use fourth-generation therapies in clinical practice. This is because issues related to human safety, such as translational challenges in CaP, as well as questions of efficacy, have not yet been fully addressed [30,31,32].

## 4. Biomaterials

Biomaterials serve as substitutes for human tissues that have failed to regenerate or heal spontaneously. These materials include allograft-based substitutes, natural and synthetic polymers, ceramics, and metals.

### 4.1. Allograft-Based Substitutes

Allograft-based substitutes, commonly known as demineralized bone matrixes (DBM), are the result of the sterilization, decellularization, and demineralization of original bone tissue. The production of DBM involves a rigorous control process to ensure the preservation of growth factors, collagen, and non-collagenous proteins that are inherent to native bone tissue [33]. Allografts typically serve as an osteoconductive scaffold with limited osteoinductive capabilities. However, due to their non-viable cellular composition, allografts do not possess any inherent osteogenic potential [34]. Empirical evidence indicates that DBM may possess superior osteoinductive qualities owing to the exposure of soluble factors consequent to demineralization. This exposure may otherwise be impeded in mineralized bone [35]. However, DBM-based bone is limited by the lack of osteoconductivity or osteogenicity, and the mechanical properties of DBM are below the optimal range [36]. According to Kwon et al., the stability and fusion rate of allografts can be increased by adjusting the cortico-cancellous composition [37]. According to Schizas et al., the fusion rate and postoperative function of 33 patients who underwent lumbar fusion using a combination of injectable DBM products and iliac crest bone did not differ significantly from those of 26 patients who received iliac crest bone alone [38].

### 4.2. Cell-Based Substitutes

Given the differentiation potential of mesenchymal stem cells/stromal cells (MSCs) towards cartilage, cell therapy has emerged as a promising avenue for tissue regeneration and repair. MSC transplantation, in particular, has demonstrated significant efficacy in this regard. The use of MSCs as seed cells can prevent or delay the need for joint replacement surgery since they immediately contribute to local repair and, through their secretory capabilities, regulate metabolism and immune function. The adaptable nature of MSCs positions them to assume pivotal roles across multiple stages of the cartilage repair process. In addition, the joint cavity’s accessibility and relatively constrained dimensions make injecting MSCs for joint conditions significantly more practical than treating systemic disorders that mandate the administration of MSCs via systemic injection. The safety and dependability of MSC therapy for OA have been demonstrated in a number of pre-clinical and clinical investigations [39,40,41]. In the field of knee surgery, cartilage salvage surgery using allogeneic umbilical-cord-blood-derived mesenchymal stem cells (CARTISTEM^®^ (Medipost Co., Ltd., Sungnam, Gyeongi-do, Republic of Korea)) has been performed for several years. In the field of spine surgery, several preclinical and clinical studies have been reported. Despite preclinical studies suggesting that MSCs can enhance sensory and motor function recovery in rats, the efficacy of MSCs in treating spinal cord injuries (SCIs) remains a subject of controversy. Clinical trials have reported improvements in American Spinal Injury Association (ASIA) sensory and motor scores following MSC administration [42]. In the single-level interbody fusion model in sheep, the utilization of osteoconductive scaffolds for delivering adult allogeneic MPCs proved to be both safe and efficacious. These findings support the potential use of allogeneic MPCs as a viable alternative to iliac crest autograft for lumbar interbody spinal fusion procedures [43]. In this ovine spine fusion model, the use of adult allogeneic mesenchymal precursor cells delivered via a hydroxyapatite:tricalcium phosphate carrier demonstrated both safety and efficacy. The findings from this preclinical study indicate that allogeneic mesenchymal precursor cells can achieve fusion efficacy similar to that of iliac crest autograft, providing a viable and safe alternative for achieving successful posterolateral spine fusion [44]. The existing evidence is currently inadequate to endorse the use of MSCs or BMA in conjunction with synthetic or allogeneic materials as a replacement or adjunct to autologous bone grafts in humans [45].

### 4.3. Natural Polymers

Collagen, fibrin, and chitosan are representative natural polymers. Natural polymers provide excellent osteoconduction but offer limited osteoinduction, osteogenicity, and mechanical properties compared with autograft bone tissue. Commercial bone graft substitutes derived from natural polymers are also available. One example of a commercially available natural polymer product is Healos^®^ (DePuy Orthopedics, Inc., Yongsan-gu, Seoul, Republic of Korea), a collagen microfiber matrix coated with HA. Prior to implantation, a bone marrow aspirate coating is recommended for Healos^®^ to provide osteogenicity, which is a limitation of natural polymers. In addition to collagen, two more natural polymers that are being looked into for bone tissue engineering applications are fibrin and chitosan. In recent years, there has been growing interest in the use of chitosan-based biomaterials for nerve tissue engineering applications in spinal cord injury repair. According to Wei Xiang et al. [46], a range of mechanisms and functions have been attributed to chitosan-based biomaterials in the promotion of SCI repair, including the facilitation of neural cell growth, guidance of nerve tissue regeneration, delivery of nerve growth factors, and serving as a vector for gene therapy. These polymers often produce fibers or foams as their final forms. Excellent osteoconduction is supplied by these structures, but they have worse osteoinduction, osteogenicity, and mechanical qualities compared to autograft tissue [47,48].

### 4.4. Synthetic Polymers

Synthetic polymer-based substitutes are diverse and widely studied, especially in the current orthopedic field. Compared with natural polymers, synthetic polymers allow for the finer control of surface chemistry, degradation kinetics, and geometry. The growing popularity of polymers can be attributed to their cost-effectiveness and broad applicability [18]. First-generation polymeric biomaterials include polyethylene (PE), polymethylmethacrylate (PMMA), and polyurethane (PU), and both first-generation and third-generation variants of these materials continue to be used today. Maitz et al. [49] provides a comprehensive review of all synthetic polymer types used in clinical medicine. Table 1 lists the advantages and disadvantages of the most commonly used synthetic polymers used in orthopedics.

#### 4.4.1. Polyethylene

PE is utilized in disc replacement, tibia insertion, and total hip and knee arthroplasty [10,60,61]. Low friction resistance, resistance to abrasion or impact, and good biocompatibility are the primary advantages of PE. The biggest disadvantage of PE is the potential for debris production due to wear over time. A specific type of PE is ultra-high-molecular-weight PE (UHMWPE). The physical, chemical, and mechanical characteristics of a polymer are determined not only by its molecular weight, but also by its microstructure. The combination of mechanical properties and excellent wear and abrasion resistance is important. UHMWPE has unique properties, such as chemical inertness, low friction, high impact strength, exceptional toughness, low density, ease of manufacturing, biocompatibility, and biostability [62,63,64]. Since its introduction in 1962, ultra-high molecular weight polyethylene (UHMWPE) has been employed in orthopedic surgery as a bearing material. It is used as a liner in acetabular cups for total hip arthroplasties, as well as in the tibial insert and patellar element for total knee arthroplasties. Furthermore, UHMWPE is utilized in intervertebral artificial disc replacements as an insertion. The objective of present-day research is to prolong the onset of oxidation by incorporating appropriate stabilizing agents that can impede the reactivity of free radical species and decelerate the oxidation mechanisms without compromising the chemical, physical, or mechanical features of the material, thus extending its durability [65,66]. One potential stabilizing compound for UHMWPE is vitamin E (alpha-tocopherol), which is a natural antioxidant present in the human body. Using vitamin E-stabilized PE and irradiation produces a cross-linked UHMWPE, known as second-generation HXLPE, which exhibits improved stability against oxidation [67]. The aforementioned material became commercially available in 2007–2008, and had a cross-link density sufficient to provide long-term oxidative stability and excellent wear performance. As a result, it can withstand extended wear while retaining its mechanical properties. The addition of vitamin E to UHMWPE may also reduce the inflammatory response to wear particles. Infection, in addition to wear and component deterioration, remains a significant obstacle to total joint replacement [68].

#### 4.4.2. Polymethylmethacrylate

PMMA is known as acrylic cement, providing superior osteointegration, good tensile properties, good flexural rigidity, and the absence of interim fibrous tissue around cemented elements [53,54]. However, PMMA is limited by the potential for microfractures, the release of cement particles (such as methacrylate monomers [MMA]) and heat, localized foreign-body reactions, and limited biological response [53,54]. The release of MMA can lead to hemodynamic effects that may include reductions in blood pressure and oxygen saturation levels [18]. PMMA is a versatile substance utilized in orthopedics, dentistry, and ophthalmology. PMMA is used in orthopedics as a permanent bone substitute to treat pathologic fractures and is also used in internal fixation plates, spinal fracture fixation, and hip arthroplasty [69,70]. The most important feature of PMMA is its ability to be shaped into the specific forms required by implants or polymerized in situ while the patient is undergoing surgery. The polymerization process lasts 6 to 7 min. When PMMA is used as a cement to secure prostheses to the bone, bone adhesion is ensured by including HA particles in the polymer, resulting in a uniform load transmission from the implant to the bone [71,72]. For fragility fractures, such as in vertebroplasty or kyphoplasty, PMMA injections provide increased stability and decreased pain [73,74,75]. The extravasation of PMMA to the spinal canal and foramen is a major potential problem that may affect the spinal nerve root [74]. In response to these concerns, there has been recent interest in using cannulated fenestrated screws in combination with PMMA cement for surgical treatment of osteoporotic vertebral fractures [76]. Recently, various bone cement cannulated fenestrated screws for osteoporosis patients have been developed. For example, The Iliad™ pedicle screw system (Medyssey Co., Ltd., Jecheon, Republic of Korea), which is a cannulated fenestrated screw system, underwent a study regarding the optimal design of cement-augmented screws to reduce cement leakage-related complications (Figure 2) [73]. The modified design of the fenestrated screw and the application of the suggested cement injection pressure resulted in minimal spinal canal leakage and easy access.

#### 4.4.3. Polyurethanes

Over the past twenty years, polyurethane (PU) biomaterials have been extensively investigated for their possible use as compliant orthopedic materials, including as bearings. PU materials are believed to work under a microelastohydrodynamic lubrication regime and have lower modulus values than UHMWPE, resulting in reduced wear [77].

Depending on the objective of the implanted device, various biomaterials can be used that each offer distinct features, including elasticity, mechanical strength, flexibility, and stiffness. The most significant property of an implanted device is the ability to mimic biological characteristics, especially bone structure. For example, PU is used clinically in vascular catheters, nerve regeneration guidance conduits, heart valves, stents, and orthopedic implants [78]. Devices implanted using PU can be transformed into completely biocompatible materials suitable for permanent implantation or biodegradable materials that can be used as scaffolds for tissue regeneration [79]. Segmented polycarbonate urethanes (PCUs), third-generation PU biomaterials, have more oxidative stability than PU. PCUs have been the subject of research as a potential bearing material for total acetabular replacement, owing to their noteworthy characteristics such as exceptional toughness, ductility, oxidation resistance, and biostability [80,81,82]. The majority of materials used for mechanical testing of orthopedic fixation devices are PUs, as chemically modified PUs have the ability to accurately simulate both the compact structure of cortical bone and the trabecular structure of spongy bone. Because human bone and polymers offer comparable mechanical performances, standardized PU foams were created for various bone densities [83]. Thompson et al. [84] tested different rigid PU foam types and found that the polymer behaves with similar elasticity to spongy bone. In studies of biomechanics, PU foams can be used as substitutes for human bones to determine the essential functional parameters (such as resistance, stability, and rigidity) of orthopedic implants. This can help to understand how the implants will perform in vivo and potentially improve their design and effectiveness [85].

#### 4.4.4. Polyetheretherketone (PEEK)

As its biocompatibility was confirmed in the 1980s, polyetheretherketone (PEEK), a variation of polyaryletherketones (PAEKs), has become more often utilized as a biomaterial for orthopedic, trauma, and spinal implants [86]. A series of high-temperature thermoplastic polymers, known as PAEKs, includes a ketone and ether functional group-interconnected aromatic backbone molecular chain. Nowadays, PEEK is utilized for femoral stems, bearing materials for hip and knee replacement, and hip resurfacing. PEEK became prevalent in the late 1990s due to its recognition as a leading high-performance thermoplastic material that could replace metal implant components, particularly in the field of trauma and orthopedics [86]. Its resistance to in vivo deterioration, including harm from lipid exposure, was a key selling point. In order to increase implant fixation, PEEK was later made available as a biomaterial for implants in April 1998. Subsequent experiments examined the interaction of PEEK with bioactive substances, such as hydroxyapatite (HA), either as a composite filler or surface coating. Ongoing research efforts enable the customization of PEEK and similar composites with various physical, mechanical, and surface characteristics to meet specific application requirements.

Brantigan promoted the use of PEEK as a biomaterial for cages after a pilot clinical research he conducted in 1989 produced positive results and 100% radiological identification of the interbody fusion. In recent studies using PEEK, hydroxyl-apatite (HA), 40% tricalcium phosphate, 60% hydroxyl-apatite, and rhBMP-2 were combined on a collagen sponge to improve and speed up bony fusion rates [87]. Comparing neet PEEK cages to an n-TiO_2_/PEEK composite, a study in 2012 by Wu et al. found that much more bone growth occurred [88,89]. One material that has piqued interest for its potential in demanding applications, such as complete disc replacement and posterior dynamic stabilization, is PEEK. However, some issues may arise with the use of PEEK, including subsidence and wear and fracture due to the weight-bearing demands placed on the implant.

#### 4.4.5. Silicon-Based Materials

Silicon-based materials are synthetic polymers with numerous medical applications and are of interest for orthopedic applications. Biocompatibility and biodurability are two characteristics of silicon-based materials that have long been recognized. Silicon-based materials are composed of silicon (Si), oxygen, and, frequently, carbon and/or hydrogen. One of the most commonly employed silicone polymers for medical applications is polydimethylsiloxane (PDMS). The synthesis conditions dictate whether PDMS is produced as a viscous fluid, a soft gel, or a hard elastomer [90]. Linear PDMS polymers are typically in the form of highly viscous liquids or gel-like jellies. Outstanding chemical stability, no toxicity, reasonable cost, and the capacity to construct them into a variety of shapes are all characteristics of silicone elastomers that make them particularly beneficial for medical applications. One result of this is the development of well-established silicone rubbers used in a variety of medical devices, including medical-grade tubing, shunts, catheters, breast implants, and penile implants, among others. A common application of silicone in the field of orthopedics is in the development of joint implants for the hand and foot, such as the silicone finger joint implants designed by Swanson [57]. Hand and foot implants of a similar design have been created. The most common kind of small joint implant is still made of silicone. Recent research on silicone metacarpophalangeal joint replacement has shown good long-term outcomes, high survival, and favorable patient reaction [91].

### 4.5. Ceramics

The increased longevity of the population has prompted the prevalent adoption of ceramic-based biomaterials as substitutes for bone grafts. The favorable characteristics of biocompatibility, high hardness, and high wear resistance make ceramic-based substitutes a viable option for bone replacement bearings [92]. Advances in the use of ceramics as bearings in arthroplasty have reduced clinical wear, reducing the risk of debris-induced osteolysis [93].

Bioceramics can be divided into three types: bio-inert (alumina, zirconia, tantalum oxide), biodegradable (calcium phosphates), and bioactive (bioactive glass). The orthopedic field frequently utilizes calcium phosphate coatings due to their mineral-like properties that aid in increasing fusion rates during the fusion phase, as well as their exceptional biocompatibility and capacity for osteointegration with living tissue. However, ceramics are limited by the lack of both osteoinductivity and osteogenicity [14]. Alumina–zirconia ceramic composites exhibit exceptional mechanical and stability qualities but are expensive to produce [94]. Orthopedic applications have incorporated calcium phosphate coatings due to their likeness to the mineral composition of bone, ability to facilitate bone growth, and advantageous biocompatibility and osseointegration with the host tissue. The lack of uniformity and limited control over thickness and surface topography in plasma sprayed calcium phosphate coatings can lead to implant inflammation if particles become detached from the coating [95]. For hard-on-hard hip bearings, silicon nitride (Si_3_N_4_) is a recent addition to the ceramic biomaterials market [96]. This material preserves its engineered mechanical characteristics while possessing exceptional surface chemistry, which enables a gradual release of silicon and nitrogen in aqueous environments. This particular feature promotes the regeneration of soft and osseous tissue while concurrently inhibiting bacterial growth. Si_3_N_4_ was found to have not only antibacterial effect but also antiviral effect. By Pezzotti et al. [59], chemical reactions occurring on the surfaces of Si_3_N_4_ have the potential to deactivate various types of single-stranded RNA viruses, regardless of whether they possess an envelope or not [97]. These advantages enable its application in a wide range of fields, both inside and outside the human body, such as orthopedics, dentistry, virology, agronomy, and environmental cleanup. Given the danger that evolving viruses and bacteria cause to human health worldwide, silicon nitride presents a viable and uncomplicated alternate strategy for combating these diseases. By changing the additive composition during manufacture, a variety of Si-based, nonoxide ceramics with variable properties that differ from those of the commonly used Al_2_O_3_ can be created. As Si_3_N_4_ has stronger physical properties than alumina, Si ceramics are typically appropriate for total joint replacement applications. There are, however, some issues as well. For instance, superficial oxidation of Si_3_N_4_ leads in a layer rich in Si oxide (SiO_2_) that is few nanometers thick and has been observed on Si_3_N_4_ and SiC surfaces. This thin coating has the potential to chip off over time, which might lead to dramatically increased third-body wear [58]. Si_3_N_4_’s biocompatibility might also rely on the ceramic formulation. Si-based ceramics have advanced in orthopedics despite these limitations.

Table 2 illustrates the benefits and drawbacks of ceramic biomaterials for implant applications.

## 5. Metal

The first implant devices were made from metals and alloys because these materials are strong and do not harm the body. The osteointegration of orthopedic implants follows a multi-step process that commences with exposure to bodily fluids, triggers an acute inflammatory response, facilitates cellular attachment to the implant surface, and ultimately promotes the generation and remodeling of new bone tissue at the implantation site [106]. Metals used in implanted devices include cobalt, nickel, iron, titanium, and zirconium. Metal alloys have been developed to achieve specific properties, such as strength, ductility, elasticity, and corrosion resistance [92]. Stainless steels, cobalt-based alloys, and titanium-based alloys are currently used in orthopedic metal-based implants [107]. The advantages and disadvantages of the most commonly used metals and alloys used in orthopedic implants are presented in Table 3.

Stainless steel 18-8, which is composed of 18% chromium and 8% nickel, is the most common alloy.

Superior corrosion resistance can be achieved by modifying the composition of stainless steel by adding more metals, especially chromium [114,115]. When chromium is present, Cr_2_O_3_ can form, creating a strong and adherent layer that prevents further oxidation [116]. Due to its low cost, stainless steel is widely used in removable orthopedic devices, such as fracture plates and hip screws [117,118]. New stainless steel–based alloys include cobalt, chromium, nickel, magnesium, and high nitrogen content [117]. Alloys of this nature are suitable for use in conjunction with PE as disc prostheses [119].

Cobalt-based alloys are stronger than stainless steel alloys [120]. Cobalt-based alloys are known to possess superior biocompatibility and corrosion resistance in comparison to stainless steel alloys, despite their higher production costs. Certain cobalt–chromium–molybdenum alloys are utilized specifically for hip prosthesis implants [16,121]. Due to its excellent abrasion resistance, this alloy type is only used in metal-to-metal devices [122].

Titanium and its alloys have gained significant popularity in orthopedic surgery owing to their exceptional biomechanical and biocompatible properties, remarkable mechanical strength, fatigue-corrosion resistance, relatively low modulus, low density, and high resistance to corrosion by aggressive physiological fluids. These superior characteristics have made them a preferred choice for implanted biomaterials over the last 60 years. Pure titanium offers low weight, excellent corrosion resistance (especially in saline solution due to the formation of an adhesive layer of TiO_2_), and the capacity to integrate firmly with bone [60,123,124,125]. The achievement of effective implant anchorage to the surrounding bone is crucially dependent on osteointegration, and, therefore, significant efforts are being made to design and optimize the surfaces of biomaterials to enhance this phenomenon [62]. The ability to integrate with bone significantly enhances long-term performance and lowers the possibility of device failure and loosening [35]. Although titanium-based alloys are particularly resistant to corrosion and are biocompatible, long-term implantation may result in the production of potentially harmful alloying components [27]. Furthermore, because the elastic modulus values of titanium-based alloys are relatively high compared to the value for bone, stress shielding is another potential complication. Due to the elevated costs of materials, the utilization of titanium-based alloys is limited to patients exhibiting hypersensitivity reactions to steel or cobalt-chromium alloys [112,126]. Incorporating inorganic bioactive layers, such as silver (Ag), copper (Cu), zinc (Zn), or cerium (Ce), is an effective approach to promote osteogenesis and osseointegration while simultaneously providing antimicrobial effects through contact-killing, release-killing, or a combination of both mechanisms on titanium surfaces [127,128]. It is possible to improve the bioactive properties of titanium and its alloys through electrochemical methods or chemical surface treatments, which can create a coating of an external substance such as apatite or bioactive glasses to enhance its performance.

Compared to typically utilized metallic materials such as titanium alloys and stainless steels, the use of magnesium would enable the creation of lightweight implants with an elastic modulus considerably more similar to bone [129]. Along with implant corrosion, this would lessen various pathological problems related to the implantation of permanent metallic components, namely the development of inflammatory wear particles and osteopenia [113,130]. The corrosion of magnesium, however, is the most significant of the above-mentioned characteristics because a biomaterial’s ability to perform requires that it maintain proper mechanical stability for a set amount of time. Recent developments in the field of metallurgy have led to the production of magnesium-based biomaterials with improved mechanical properties and higher corrosion resistance. As a result, an increasing number of surgeons are reconsidering the potential clinical applications of biodegradable magnesium alloys. Orthopedic devices and implants made of magnesium or its alloys have been evaluated for the treatment of bone flaps and fractures. For example, Zhao et al. [131] used specially created high purity Mg screws to fix vascularized bone flaps in patients with association of research circulation osseous stage II/III osteonecrosis in the femoral head (ONFH). Patients who received fixation with Mg screws showed significantly improved treatment outcomes in the Harris hip score and bone flap displacement, as confirmed by radiographic imaging throughout the 12-month follow-up period.

Metallic implants are still in use and are constantly evolving, even though they are first-generation devices. Metal alloys can undergo various surface treatments to achieve specific surface characteristics such as roughness, wettability, and electrostatic charge. These factors play a crucial role in determining the quality of implant anchorage in bone [117].

## 6. Current Research Focus

Current studies are focused on developing new materials and surface modification techniques to improve orthopedic implants. The goal is to create a new generation of implants with better integration and bone healing, achieved through the development of composite coatings that closely mimic the structure of human bone. The field of biomedical nanotechnology has recently gained significant attention in this regard. Luo et al. [132] demonstrated that cerium oxide (CeO_2_) nanoparticles (NPs) improve matrix mineralization and boost osteogenic gene expression in MC3T3-E1 murine osteoblast precursors, and CeO_2_ NPs have great potential in the orthopedics field [133,134]. Samanta et al. [135] have shown that gold NPs exhibit antimicrobial properties. Because silver NPs offer antimicrobial activity and are smaller than other NPS, which have deleterious effects on bone-forming cells in vitro, Castiglioni et al. [136] advocated using silver NPs to prevent infections in orthopedic implants. These findings highlight the potential beneficial effects of nanomedicine, including enhancing the efficiency of orthopedic materials. Despite the swift progress made in 3D-printing technology, the use of 3D-printed orthopedic implants remains largely unexplored. In particular, there is a need for more in-depth research on the application of polymers, which can be used in various forms such as gels, filaments, and solutions, to create 3D-printed bone substitutes using techniques such as direct ink writing, fused deposition modeling, and stereolithography [137]. In orthopedics, 3D-printed materials can be used to generate life-size anatomical models, implants, and prostheses [138]. Pedicle screw fixation has a higher rate of complications among older patients who have low bone quality relative to younger patients. These complications can lead to loosening, pullout, and screw migration following surgical treatment for osteoporotic vertebral fractures [139]. The advent of 3D-printed materials, such as pedicle screws or interbody cages, has allowed for easier surgical treatments. When using 3D-printed materials, the incidence of complications, such as cement leakage, was also reduced among patients who previously experienced surgical difficulties (Figure 3) [73,140]. The shape and height of interbody cages are important for correcting the sagittal alignment in degenerative spine diseases, including in osteoporotic patients [141]. Therefore, the use of patient-specific interbody cages can result in a better surgical prognosis.

## 7. Limitation

This article deviates from the current basic science level since it is written from the perspective of clinical spine surgeons. In fact, biomimetric biomaterials known as fourth-generation biomaterials are not used clinically, and many other biomaterials are not used clinically due to problems such as biostability and less clinical evidence of high quality. In addition, 3D printing devices such as cages are reporting less favorable results which is different from the in vitro or bioexperimental results [142]. However, clinicians still need to understand the past, present, and future of biomaterials because they are the actual users of biomaterials, so they need to know a deeper and more specific area of biomaterials.

Moreover, we experienced an unexpected gap between the basic scientists and clinicians in the development of improved design of fenestrated pedicle screws for cement augmentation. The most biomechanically sound design of fenestrated screws could not be applied in real osteoporotic patients in actual clinical practice, due to high risk of cement leakage into the spine canal, which could result in neurologic compromise. This gap between safety and efficiency will be a common phenomenon for all other biomaterial and biotechnologic devices. To bridge this gap and develop clinically applicable biomaterials, numerous studies on biomaterials in a physiological environment, as well as improved interactions between clinicians and basic scientists, would be necessary. Rather than comprehensively dealing with biomaterials at the basic science level, this review deals with the view of biomaterials from the perspective of a clinical spinal surgeon.

## 8. Conclusions

This review explores the basic properties of biomaterials and metals currently being used in orthopedics. Orthopedic problems that can be encountered include fractures, broken joints, and trauma to bone that is already diseased [117]. The use of prostheses or implantation devices is necessary to restore normal bone or joint function. To ensure that the patient experiences a full recovery, the implant material must have certain qualities. Currently, the majority of orthopedic implants in use are made of metals due to their widespread availability, durability, and well-established manufacturing processes. However, there are limitations in terms of potential enhancements through structural or surface modifications, as well as reinforcement with more elastic and durable materials. In the future, polymers and composites may prove to be more suitable, as there are several candidates that surpass metals in terms of elasticity and durability, without the risk of sensitization and intoxication. New varieties of polymers can successfully replace metal alloys, which are still commonly used. The most recent generation of synthetic materials based on biocompatible polymers offers various options for replicating anatomical structures and are capable of being absorbed over time or inducing specific responses from the biological environment. PUs are the best materials for mechanically testing orthopedic devices. Additionally, we suggest that the application of nanotechnology is crucial to improving the effectiveness of orthopedic implants. Finally, synthetic materials make it possible to simulate the behavior of an implant device, improving the likelihood of successful surgical implantation procedures. Orthopedic materials will continue to be developed, aiming to lower implant costs, uphold patient safety, improve surgical methods, and lower the risk of infection.

## Figures and Tables

**Figure 1 materials-16-03633-f001:**
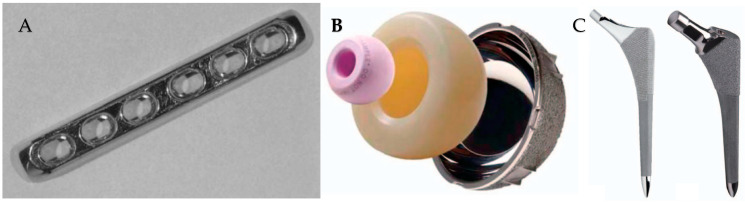
(**A**) Titanium alloy plate for internal fixation of fracture. (**B**) Pink spherical structure is ceramic head for total hip arthroplasty. (**C**) Various type of titanium alloy stem for total hip arthroplasty [2,3].

**Figure 2 materials-16-03633-f002:**
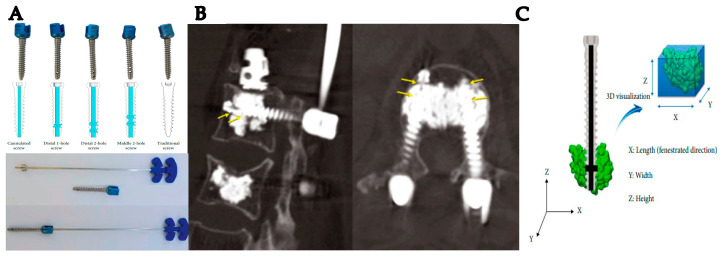
(**A**) Cement screw design and cement injector in the Iliad™ pedicle screw system. (**B**) The guide arrow indicate the side hole of fenestrated screw in clinical computed tomography. (**C**) Three-dimensional analyses of cement distribution [73].

**Figure 3 materials-16-03633-f003:**
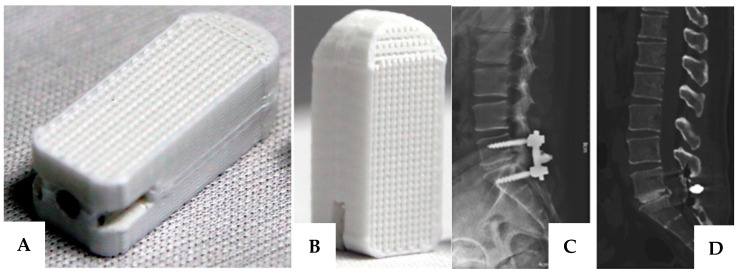
(**A**,**B**) Lateral view and vertical view of 3D printed PCL/β-TCP cage, (**C**,**D**) Posterolateral interbody fusion with a 3D printed cage at L5/S1 [140].

**Table 1 materials-16-03633-t001:** The advantages and disadvantages of the most commonly used synthetic polymers used in orthopedics.

S. No.	Polymer Base	Advantages	Disadvantages	Reference
1	Polyethylene (PE)	Very ductile and has good impact strength, low friction Coefficient, low fracture toughness	Low strength and hardness, Wear can cause debris over prolonged periods,	[50,51]
2	Ultra-High-Molecular-Weight (UHMWPE)	Superior mechanical properties than PE; enhanced modulus; higher creep resistance	Releases debris; adverse tissue biological reactions; osteolysis or bone loss leading to implant loosening	[52]
3	Polymethylmethacrylate (PMMA)	Superior osteointegration; The lack of intermediate fibrous tissue surrounding the cemented components	Faces microfractures; releases cement particles; limited biological response	[53,54]
4	Polyurethane (PU)	Inexpensive; offers diverse and specific properties; mimics bone structures	Emission of bisphenol A that exhibits estrogenic activity	[55]
5	Polyetheretherketone(PEEK)	Biocompatibility, biostability, less weight, good mechanical properties, stable at high temperatures.	Difficult and expensive manufacturing process, low bioactivity.	[56]
6	Silicon	High biocompatibility, high biodurability. Antibacterial and antiviral effect	High corrosive wear rate	[57,58,59]

**Table 2 materials-16-03633-t002:** The advantages and disadvantages of ceramic biomaterials for implant applications.

S. No.	Ceramic Based		Advantages	Disadvantages	Reference
1	Bioinert	Zirconia	Reduces wear and risk of dislocation, biocompatibility	Hydrothermal aging, phase transition, costly, and susceptible to slow crack growth which can cause delayed failure.	[98]
		Alumina	Good tribological behavior	Poor fracture toughness, high cost, and squeaky issue	[99,100]
		Tantalum oxide	Biocorrosion and wear resistance, low ion release	Dislocation, instability	[101]
2	Biodegradable	Calcium phosphates	Trabecular bone-like mechanical characteristics	Low tensile strength and unstable at high temperatures	[102]
		Hydroxyapatite	Promotes bone growth, cheap, high osteointegration properties	Delamination and abrasion wear.	[103]
3	Bioactive	Bioactive glass	Accelerated bone development, improved fixing, improved cell growth, and improved stability	Low bending and fatigue strength, low fracture toughness, brittle nature, difficult to produce, and low ductility	[100]
		Silicon nitride	High fracture toughness and strength; low wear properties.	Brittleness; low energy dissipation	[104,105]

**Table 3 materials-16-03633-t003:** The advantages and disadvantages of the most commonly used metals and alloys.

S. No.	Metal Base	Advantages	Disadvantages	Reference
1	Stainless steel	Inexpensive; high wear resistance; good fatigue resistance	Crevice and pitting corrosion; stress shielding effect; nickel (Ni) and chromium (Cr) allergy	[108,109]
2	Cobalt–Cr-based alloys	High corrosion resistance with minimal susceptibility; high wear resistance; biocompatibility; load-bearing materials	Early loosening rate; limited use; Ni and Cr allergy	[109,110]
3	Titanium (Ti)	Superior mechanical properties; lightweight; biocompatibility	Low shear strength; poor tribological properties; expensive	[109,111]
4	Ti–Nb (Niobium)–Ta (Tantalum)–Zr (Zirconium) alloy	Favorable mechanical properties; excellent biological characteristics; superior wear resistance; biocompatibility; great corrosion resistance,	Low strength; expensive	[112]
5	Mg alloys	Low density, degradable	Low corrosion resistance; limited understanding of the tissue response	[113]

## Data Availability

Data reported were taken from papers included in the references.

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
