# Peer review of "The Clinical Use of Osteobiologic and Metallic Biomaterials in Orthopedic Surgery: The Present and the Future"

_materials, 2023, doi:10.3390/ma16103633_

Round 1
Reviewer 1 Report (Previous Reviewer 1)
· The introduction is very poor, you can mention a short comparation of the classes of biomaterials (advantages, disadvantages etc.). The following references are suggested: [1] https://doi.org/10.3390/bioengineering9110686; [2] DOI10.3390/mi12121447
· You can add a figure with applications in the human body.
· On metallic implants, mention the Magnesium alloys.
· Add more recent references.
· Make an image to put in the introduction to highlight the overall of the paper.
· In the Introduction section, the authors cited the specific results of previous research and cited them adequately. However, they did not mention their shortcomings in previous research. In the Introduction section, the penultimate paragraph should contain common features of previous research. The shortcomings of previous research should also be pointed out, in general.
· In the Introduction section, the last paragraph should contain the scientific contribution and scientific hypotheses of your research. Complete, further elaborate the scientific contribution and scientific hypotheses of your research. Be explicit. In addition to the goal of the research (which was written), the novelty in the context of the scientific contribution should be pointed out. Scientific contributions should be written based on the shortcomings of previous research in the literature. In this way, the authors will better emphasize novelty and scientific soundness.
· Discuss about mechanical properties of orthopedic implants.
Author Response
Thank you for giving me the opportunity to submit a revised draft of my manuscript titled ‘The clinical use of osteobiologic and metallic biomaterials in orthopedic surgery; the present and the future.’ to Materials appreciate the time and effort that you and the reviewers have dedicated to providing your valuable feedback on my manuscript. We are grateful to the reviewers for their insightful comments on my paper. We have been able to incorporate changes to reflect most of the suggestions provided by the reviewers. We have highlighted the changes within the manuscript.
Here is a point-by-point response to the reviewers’ comments and concerns.

Reviewer 2 Report (Previous Reviewer 3)
The manuscript has increased once this is a resubmission being now able to be published.
Still I suppose the references are not correctly inserted in the MDPI format.
Author Response
Thank you for giving me the opportunity to submit a revised draft of my manuscript titled ‘The clinical use of osteobiologic and metallic biomaterials in orthopedic surgery; the present and the future.’ to Materials appreciate the time and effort that you and the reviewers have dedicated to providing your valuable feedback on my manuscript. We are grateful to the reviewers for their insightful comments on my paper. We have been able to incorporate changes to reflect most of the suggestions provided by the reviewers. We have highlighted the changes within the manuscript.
Here is a point-by-point response to the reviewers’ comments and concerns.

Reviewer 3 Report (Previous Reviewer 2)
I thank the authors for this new version of this paper. Due to the improved introduction, the aim of the study is much more understandable and relevant.
To m opinion, the fact that this paper is an overview of the currently used biomaterials in orthopaedic surgery has to be even more emphasized (in the abstract, introduction, and conclusion). According to me, this is the main point for such overview addressed to an engineering and scientists community.
Please find some more detailed comments below:
General: Please be careful to typographical mistakes within the text.
1. Introduction
This introduction is better, to my opinion. It really shows that this paper is an overview, and that such overview is interesting for both clinician and researchers. I believe that the authors can even more emphasize and state that they propose an overview for biomaterials that are used in clinics (if this is the case for this overview).
Indeed, this introduction lacks a state of the art. There have been numerous reviews on bone biomaterials. Why this one is needed compared to the others? To my opinion, it is interesting because it relates biomaterials that are used in clinics and not just only investigated in laboratories (this a very important point for researchers!).
In regard of this, I also suggest to the authors to provide their clinician opinion on what is a “good” biomaterial to be used in clinics by clinicians (if there is one). Again, the reader of the Materials Journal are mostly materials scientists, mostly focusing on the manufacturing process, on the technological advanced, … and I strongly believe that this may be of interest for them to understand the clinician point of view that is a criterion that may be consider during the engineering step.
I believe, as it is a journal for materials scientists, please define what the authors consider as orthopedic surgery. This is a vast field and I believe a definition is needed so as everything is clear in the following. For example, the authors talk about bone, which is the most obvious one when considering orthopedic, but also cartilage, SCI, …
2. Osteobiology
Lines 63 – 66: Bone remodeling can occur even with a permanent prosthesis. Besides, authors talk about tissue engineering constructs while such biomaterials are very specific to the field of tissue engineering (in vitro tissue synthesis).
Lines 66 – 67: I believe the authors should explain why this is a determinant criterion.
3. The generation of biomaterials
Lines 81 – 84: This sentence presents some mistakes.
Lines 89 – 90: Please add references to this statement.
Lines 95 – 97: What does this statement mean? How resorbable material can maintain its mechanical properties? The mechanical properties of the first-generation biomaterials decrease with time?
Lines 97 – 98: Please add a reference for this statement.
Lines 98 – 100: Do the authors mean that, in clinics, only those resorbable polymer-based materials are used? If so, please precise it.
Lines 104 – 105: What is a bioabsorbable materials?
Lines 105 – 107: I believe the authors talk about bioactive glasses? Bioactivity is not only the capacity to precipitate HAP (see for example DOI: 10.1016/j.dental.2020.03.020).
Lines 108: Two thirds of what?
Lines 97 – 110: This new paragraph is not so easy to read. It lists the different application for these different 2nd generation biomaterials without any transition between them, what makes it complicated to understand their relationships. I believe that the point for these 2nd generation biomaterial is that they interact with their biological environment, compared to the first-generation biomaterials. This is not enough highlighted.
4. Biomaterials
Lines 180 – 195: This paragraph still does not correspond to a cell-based substitute but to bioactive molecule delivery. These biomolecules do have an effect on the cells, but they are not cells; I believe this kind of biomaterials may be referred as “drug delivery system” or simply “delivery system”.
Line 207: Please define SCI.
Line 327: Silicon-based are not only polymers. Silicone is a polymer, but Si3N4 is a ceramic. It should not be in the Synthetic polymer section, or at least leave the silicone in the polymer section, and move the Si3N4 in the ceramics section.
Lines 367: HA is a kind of calcium phosphate.
Lines 396 – 398: Could the authors provide a reference for this statement?
Line 437: pleas modify “current study” into “current studies”.
Line 471: Please modify within the text, all the authors should have contributed to this paper.
Line 469: what is needed in response to these limitations? Better experimental or numerical modelling to improve the understanding of the biomaterials in an environment that simulate the physiological environment? A enhanced interaction between engineers and clinicians?
Tables: Could the authors add a column with some main examples of clinical use of the different materials listed?
Author Response
Thank you for giving me the opportunity to submit a revised draft of my manuscript titled ‘The clinical use of osteobiologic and metallic biomaterials in orthopedic surgery; the present and the future.’ to Materials appreciate the time and effort that you and the reviewers have dedicated to providing your valuable feedback on my manuscript. We are grateful to the reviewers for their insightful comments on my paper. We have been able to incorporate changes to reflect most of the suggestions provided by the reviewers. We have highlighted the changes within the manuscript.
Here is a point-by-point response to the reviewers’ comments and concerns.

Round 2
Reviewer 1 Report (Previous Reviewer 1)
Paper was improved, can be published.
Author Response
Thank you for giving me the opportunity to submit a revised draft of my manuscript titled ‘The clinical use of osteobiologic and metallic biomaterials in orthopedic surgery; the present and the future.’ to Materials appreciate the time and effort that you and the reviewers have dedicated to providing your valuable feedback on my manuscript. We are grateful to the reviewers for their insightful comments on my paper. We have been able to incorporate changes to reflect most of the suggestions provided by the reviewers. We have highlighted the changes within the manuscript.

Reviewer 3 Report (Previous Reviewer 2)
I thank the authors for this new version of the paper.
I have some minor comments to be considered.
Abstract: I believe abstract should state that this overview talks about biomaterials that are used in clinics, and that scientists – clinicians discussions are necessary in the future.
Line 54: Is it “physician” or “clinician”?
Line 71: Stress shielding does not result in osteoporosis but in bad remodeling. Osteoporosis is a disease associating with the pathological imbalance between bone resorption and formation. The bad remodeling due to stress shielding is not characterized by a pathological imbalance, but by an imbalance due to bad biomechanical environment. Further, there is not only stress shielding, but also reactions to released implants particles that affect bone homeostasis.
Line 147: What is “bioreabsorbable”?
Sincerly,
Author Response
Thank you for giving me the opportunity to submit a revised draft of my manuscript titled ‘The clinical use of osteobiologic and metallic biomaterials in orthopedic surgery; the present and the future.’ to Materials appreciate the time and effort that you and the reviewers have dedicated to providing your valuable feedback on my manuscript. We are grateful to the reviewers for their insightful comments on my paper. We have been able to incorporate changes to reflect most of the suggestions provided by the reviewers. We have highlighted the changes within the manuscript.

This manuscript is a resubmission of an earlier submission. The following is a list of the peer review reports and author responses from that submission.
Round 1
Reviewer 1 Report
· The introduction is very poor, you can mention a short comparation of the classes of biomaterials (advantages, disadvantages etc.). The following references are suggested: [1] https://doi.org/10.3390/bioengineering9110686; [2] DOI10.3390/mi12121447
· You can add a figure with applications in the human body
· On metallic implants, you do not mention Magnesium
· Add more recent references
· In the Introduction section, the authors cited the specific results of previous research and cited them adequately. However, they did not mention their shortcomings in previous research. In the Introduction section, the penultimate paragraph should contain common features of previous research. The shortcomings of previous research should also be pointed out, in general.
· In the Introduction section, the last paragraph should contain the scientific contribution and scientific hypotheses of your research. Complete, further elaborate the scientific contribution and scientific hypotheses of your research. Be explicit. In addition to the goal of the research (which was written), the novelty in the context of the scientific contribution should be pointed out. Scientific contributions should be written based on the shortcomings of previous research in the literature. In this way, the authors will better emphasize novelty and scientific soundness.
· Discuss about mechanical properties of orthopedic implants
Reviewer 2 Report
I thank the authors for sharing their work reviewing the different biomaterials used in orthopaedic surgery. It can be seen that an important bibliographical work has been done.
Still, the paper is very confusing, being mostly descriptive, listing the different materials used in orthopaedic surgery without any real objective. The article looks like a Systematic review but without the rules used for a systematic review. If the authors want to publish a systematic review, I strongly suggest to refer to the PRISMA website. Otherwise, the authors should clearly provide a critical review, not just a descriptive one, specially from a clinical point of view.
I understand that the author is a spine surgeon. I thus suggest to focus the current review on spine surgery. In that way, the authors might be more able to provide the current limitations of the biomaterials used in spine surgery and to provide a relevant problematic and objective for this review.
Furthermore, I believe that there already are some review articles regarding the biomaterials used for bone repair (some are referenced in the current paper), what limit the novelty of the present article.
Still, I strongly believe that surgeons’ point of view has to be considered. For example, the authors provide table with advantages and disadvantages of the different biomaterials. But those advantages/disadvantages are only base on the materials properties. What about the handling of such materials by the surgeon? Or the clinical outcomes (if any)? The authors could also provide a descriptive of the vertebra? And what tissue might be replaced in the case of trauma or pathologies. Such data might be relevant for Materials readers who are mostly Materials Scientists.
To me, the limitations highlighted in the “limitation” paragraph are the one that need to be used and discussed all along the paper. There is a lack of discussion between researchers and clinicians, then materials scientists and engineers develop biomaterials that may not be used in clinical practices.
But the Materials journal is more an “engineering journal”, in contrast with a clinical journal. The authors thus have to be very accurate when talking about processing of biomaterials. Still, there are several inaccuracies within the text. In addition to the lack of objective and the descriptive nature of the current paper, this makes the paper not relevant, in its current state, for a further publication in the Materials journal. I strongly suggest the authors to talk with their clinical knowledge to state the limit of currently developed biomaterials.
Introduction:
The introduction is very poor. The objective is not justified? Why such review is needed?
The generation of biomaterials
Line 63: This contrasts with what is state in the paper. How a biomaterial has to be inert but promote osteobiology, osteoinduction, and osteoconduction? The bioinertness was may be wanted back in the past, but not currently.
Lines 64 – 66: Does the reactivity of the biomaterial only depend on its degradation?
Lines 73-75: Does the formation of this fibrous tissue is due to the bioinertness or stiffness of the biomaterial? If not, I do not understand the relation between this sentence and the one before, and an explanation of why such fibrous tissue is formed might be relevant.
Lines 76 – 78: In ref 14, the authors talk about biodegradable biomaterials, but not bioabsorbable materials. What is the link between bioabsorbability/bioactivity and the fact that the biomaterials maintain their mechanical properties?
Lines 79 – 81: Bioactive materials do not produce apatite but promote apatite formation in specific conditions.
This paragraph is a bit confusing, and does not provide any relevant information. At the beginning, the authors state that there are 3 types of materials, but four are finally given. Furthermore, it is very descriptive, and does not provide any interpretation by the authors.
Is the following of the text focus on smart biomaterials? If yes, please precise it. If
Biomaterials
Line 98: What do the authors consider as “bone fusion”?
Cell-based substitutes (lines 116 – 131): Here the title of the section highlights cell-based strategies, but the text only deals with growth factors. This is not the same thing.
Table 1: Has PE a high or low fracture toughness?
Table 1: What does superior mechanical properties mean?
Line 187: UHMWPE also releases particles.
Lines 213 – 215: I do not understand this sentence. Stiffness and elasticity are quite the same.
Lines 258: Silicon is chemical element while silicone is a polymer. The word Silicons is wrong. Do the authors talk about silicon-based materials or about silicones? It has to be noticed that Si3N4 is not a silicone, but a silicon nitride.
Table 2: Why the risk of dislocation is an advantage for Zirconia? Why slow crack growth is a disadvantage?
Table 2: What does “higher mechanical characteristics” mean for Alumina?
Table 2: Why high stiffness is an advantage?
Table 2 is difficult to read, it is not understandable which material belongs to which type of material (bioinert, biodegradable, bioactive).
Table 3: high elastic modulus as an advantage and stress shielding as a disadvantage for the stainless steel is a bit contradictory, as stress shielding occurs because of high elastic modulus.
Limitation:
It is indeed an important limitation to not provide a sufficient science level when submitting a paper. Regarding to this paragraph, and from the surgeon point of view, why provide information on biomaterials that are not used in clinics is relevant?
The second section of this paragraph is interesting and important (Lines 407 – 414). I believe that According to me, this is a major point a Surgeon can provide in such Materials Science journal: aware the materials scientists and engineers on the clinical practice, and on why some biomaterials cannot be used in real clinical practices.
Reviewer 3 Report
The review “The clinical use of osteobiologic and metallic biomaterials in 2 orthopedic surgery: the present and the future” is relevant and properly written.
However, I suggest two minor corrections:
Throughout the all document the bibliography is not properly formatted.
Page 2, Line 63: The sentence “most important property of an orthopedic device is biological inertness” does not make sense in this context. Authors describe second, third and fourth generation of biomaterials. All of them are based in non-inert approaches. The only biomaterials that are supposed to be bio-inert were the first generation ones. Please rephrase this part.
After these changes the manuscript is publishable.